# Short-term plasticity in the human visual thalamus

**Jan W Kurzawski[1,2], Claudia Lunghi[1,3], Laura Biagi[2], Michela Tosetti[2,4], Maria Concetta Morrone[1,2], Paola Binda[1]***

[1]University of Pisa, Pisa, Italy; [2]IRCCS Stella Maris, Pisa, Italy; [3]Laboratoire des systèmes perceptifs, Département d'études cognitives, École normale supérieure, PSL Research University, Paris, France; [4]IMAGO7 Foundation, Pisa, Italy

**Abstract** While there is evidence that the visual cortex retains a potential for plasticity in adulthood, less is known about the subcortical stages of visual processing. Here, we asked whether short-term ocular dominance plasticity affects the human visual thalamus. We addressed this question in normally sighted adult humans, using ultra-high field (7T) magnetic resonance imaging combined with the paradigm of short-term monocular deprivation. With this approach, we previously demonstrated transient shifts of perceptual eye dominance and ocular dominance in visual cortex (Binda et al., 2018). Here, we report evidence for short-term plasticity in the ventral division of the pulvinar (vPulv), where the deprived eye representation was enhanced over the nondeprived eye. This vPulv plasticity was similar as previously seen in visual cortex and it was correlated with the ocular dominance shift measured behaviorally. In contrast, there was no effect of monocular deprivation in two adjacent thalamic regions: dorsal pulvinar and Lateral Geniculate Nucleus. We conclude that the visual thalamus retains potential for short-term plasticity in adulthood; the plasticity effect differs across thalamic subregions, possibly reflecting differences in their corticofugal connectivity.

*For correspondence:
paola1binda@gmail.com

**Competing interest:** The authors declare that no competing interests exist.

## Editor's evaluation

This paper documents short-term plasticity in a subcortical region, the ventral division of the pulvinar, following monocular deprivation in adult humans. This finding advances the understanding the mechanisms of short-term visual plasticity which until now has been thought to be confined to visual cortex. The results will be of interest to neuroscientists interested in brain plasticity and has potentially broad implications for the understanding of visual processing and visual disorders.

## Introduction

A classic paradigm for probing brain plasticity is monocular deprivation. During development, patching one eye for several days weakens the cortical representation of the deprived eye producing a stable change of ocular dominance columns in primary visual cortex (*Hensch and Quinlan, 2018*; *Wiesel and Hubel, 1963*; *Wiesel and Hubel, 1965*). In adult humans, a much shorter period of eye patching (about 2 hr) produces a paradoxical enhancement of the deprived eye signal (*Bai et al., 2017*; *Binda and Lunghi, 2017*; *Castaldi et al., 2020*; *Chadnova et al., 2017*; *Lunghi et al., 2015a*; *Lunghi et al., 2011*; *Lunghi et al., 2013*; *Lunghi et al., 2015b*; *Lunghi and Sale, 2015c*; *Lunghi et al., 2019*; *Lyu et al., 2020*; *Min et al., 2018*; *Schwenk et al., 2020*; *Wang et al., 2020*; *Zhou et al., 2015*; *Zhou et al., 2013*; *Zhou et al., 2014*) that was interpreted as a form of homeostatic plasticity (*Turrigiano, 2012*). Recently, we explored the neural underpinnings of this effect using ultra-high field functional magnetic resonance imaging (fMRI). Although our technique did not directly

measure ocular dominance columns, we were able to detect short-term plasticity effects in primary visual cortex V1 that were compatible with a change in ocular drive (*Binda et al., 2018*).

While ocular dominance plasticity has been thoroughly investigated in the visual cortex, less is known about its effects on subcortical visual processing.

Multiple nuclei in the thalamus are involved in processing visual signals and relaying them to the cortex. Lateral Geniculate Nucleus (LGN) is the main retinorecipient thalamic nucleus and the main source of feedforward signals to V1 (*Blasdel and Lund, 1983*; *Hendrickson et al., 1978*; *Hubel and Wiesel, 1972*). LGN cells are largely monocular (*Casagrande and Boyd, 1996*), organized in layers based on cell type (magno- parvo-cellular layers with intermixed konio-cells) and eye-of-origin (ipsi- and contralateral). However, there are indications that interocular interactions are possible in LGN (*Dougherty et al., 2021*; *Zeater et al., 2015*), either due to local interthalamic circuits (*Dougherty et al., 2019*) or to the large contingent of corticothalamic fibers that feedback into LGN from striate and extrastriate visual areas (*Adusei et al., 2021*; *Briggs and Usrey, 2011*; *Fitzpatrick et al., 2009*; *Hendrickson et al., 1978*; *Lund et al., 1975*). Whichever their origin, these interocular interactions could account for the oscillations of LGN BOLD responses during binocular rivalry, shown in seminal work by *Wunderlich et al., 2005* and *Haynes et al., 2005*.

Adjacent to LGN, the pulvinar is the largest thalamic nucleus displaying visual responses. Although a small inferior portion of the pulvinar receives a contingent of fibers from the retina and the superior colliculus (*Kwan et al., 2019*), the bulk of pulvinar input is binocular (*Bender, 1982*) and comes from the cortex, with which it is bidirectionally connected. The resulting cortico-pulvinar-cortical loops could participate in visual information processing (*Kaas and Lyon, 2007*; *Purushothaman et al., 2012*; *Saalmann and Kastner, 2011*; *Sherman and Guillery, 2002*; *Shipp, 2004*; *Zhou et al., 2016*) by regulating key parameters of visual cortical function such as gain and intracortical competition (e.g., *Saalmann et al., 2012*). The pulvinar may be further subdivided in subnuclei, but their identification in the in vivo human anatomy is problematic and MR studies often simplify the internal organization of the pulvinar into few subdivisions along the dorsoventral axis and/or the mediolateral direction (*Arcaro et al., 2015*; *DeSimone et al., 2015*; *Schneider, 2011*). Functional connectivity analyses clearly distinguish two subregions within the pulvinar (*Arcaro et al., 2018*): ventral (vPulv) and dorsal (dPulv). The ventral region vPulv is primarily connected with the occipital cortex, particularly with areas in the ventral pathway – in line with results in other primates (*Kaas and Lyon, 2007*). Coherently, vPulv is reliably activated by visual stimulation and follows perceptual oscillations (e.g., during binocular rivalry) even when it is presented to a passive observer (*Wilke et al., 2009*). The dPulv instead, is more strongly linked with parietal and frontal cortex and its responses are strongly modulated by the cognitive and attentional demands of the task (*Fiebelkorn et al., 2019*; *Kaas and Lyon, 2007*). Like for vPulv, also the activity of dPulv follows the perceptual alternation during binocular rivalry, but only when an active reporting task is performed, not in passive-viewing conditions (*Wilke et al., 2009*).

In previous studies, both LGN and pulvinar have been associated with plasticity. In humans, there are indications that LGN can shift function following sensory deprivation or restoration (*Levine et al., 2020*; *Castaldi et al., 2016*). In rodents, ocular dominance plasticity of LGN neurons was recently reported (*Huh et al., 2020*; *Jaepel et al., 2017*; *Sommeijer et al., 2017*), including a form of ocular dominance plasticity during development, leaving open the possibility that monocular deprivation effects may be present at the level of LGN even in the adult human. The pulvinar has also been implicated in developmental plasticity of the visual system (*Bourne and Morrone, 2017*; *Bridge et al., 2016*), particularly its inferior retinorecipient portion. Early in development, this region of the pulvinar relays retinal information to visual cortical area MT. During maturation, this connection is usually lost; however, it is preserved when LGN-V1 projections are lesioned (*Warner et al., 2015*), suggesting that this pulvinar subregion has a plasticity potential that may support residual visual abilities in these patients (*Kiper et al., 2002*; *Tinelli et al., 2013*). Thus, in principle, both LGN and pulvinar may support reorganization of visual processing; however, no previous study has tested the potential for short-term plasticity – specifically in response to short-term monocular deprivation – in the adult human thalamus.

One way in which short-term monocular deprivation could affect ocular dominance is by changing the automatic regulation of neuronal gain in monocular neurons and inducing an adaptation-like modulation (*Başgöze et al., 2018*); in this model, monocular deprivation effects could be seen in LGN.

We and others have disfavored this hypothesis and suggested that monocular deprivation effects could arise from interocular interactions (*Lunghi et al., 2013*; *Lunghi et al., 2015b*; *Zhou et al., 2013*). Even in this scenario, LGN could in principle show these effects, either because they arise through local interthalamic circuitry, or because they arise in the cortex and are inherited by LGN via cortical feedback pathways.

A similar reasoning could apply to responses in the pulvinar. As the majority of pulvinar cells receive binocular input (*Bender, 1982*), deprivation effects – if measurable at this level – would be most likely inherited from the visual cortex, but they could also arise within the pulvinar, in its small retinorecipient portion.

To test these hypotheses, we measured the plasticity response of LGN and pulvinar to short-term monocular deprivation in normally sighted human adults. We did so by gathering BOLD responses to passively viewed monocular stimuli, delivered before and after 2 hr eye patching. Mapping thalamic nuclei with MRI is notoriously difficult due to the low signal-to-noise ratio (SNR) and the small size of these structures. We overcame these limitations using ultra-high field (7 Tesla) fMRI and by extracting BOLD responses in selected regions of interest (ROIs), which were independently defined (based on the Natural Scenes Dataset [NSD], *Allen et al., 2022*).

## Results

In 18 adult participants with normal vision, we measured 7T BOLD responses to monocular visual stimulation (bandpass filtered noise, refreshing 8 times per second and presented in blocks of 9 s, followed by 12-s interstimulus intervals), delivered before and after 2 hr of eye patching (experimental design is shown in *Figure 1A*). We previously analyzed responses in visual cortical areas (*Binda et al., 2018*); here, we focused on responses in the visual thalamus. Pooling data across participants, after aligning them to the MNI template (*Avants et al., 2008*), we found that visual responses within the thalamus clustered in two foci (*Figure 1B*) that match two independently defined ROIs (green and blue outlines in *Figure 1B*): LGN and vPulv, obtained from the NSD (*Guest et al., 2021*). The third ROI, the mid-Pulv (magenta line in *Figure 1B*), failed to respond to our visual stimuli, which was expected since the stimuli were delivered passively and this region is primarily driven by images supporting execution of an active task (*Wilke et al., 2009*).

*Figure 1C* shows the temporal dynamics of the average BOLD responses extracted from these independently defined ROIs. Responses in dPulv were almost absent, showing only a weakly negative modulation during stimulus presentation, whereas reliable BOLD responses were observed in both LGN and vPulv. Although clearly weaker than previously measured in V1 (were signals peaked at about 2.5% at 9 s from stimulus onset; *Binda et al., 2018*), these were reliably larger than 0 at all points between 3 s after stimulus onset to 3 s after its offset (all $t(17) > 4.30$ and $p < 0.01$). Response dynamics was faster than in V1 (the peak response in LGN and vPulv occurred around 6 s from stimulus onset, 1TR earlier than in V1), as previously reported (*Lewis et al., 2018*). It was also slightly faster in LGN than in vPulv.

Given these differences in the response profile, we opted to quantify BOLD response amplitudes with an approach that makes minimal assumptions on temporal dynamics. Since the visual stimulus was a periodic alternation of stimulus contrast ON/OFF (ignoring variations in stimulus spatial scale that is not relevant here, see methods), visually evoked responses could be studied by Fourier analyses of the fMRI time series, taking the amplitude and phase at the stimulus fundamental frequency to estimate response magnitude and its delay (*Figure 1—figure supplement 1*; note that analyses based on general linear modeling (GLM) and event-related averaging produced the same pattern of results, as detailed below).

*Figure 1D* shows a polar plot of these measures, separately for each participant and ROI (see also *Figure 1—figure supplement 1C* for averages across participants), confirming the similar though slightly faster responses in LGN and vPulv and the very small responses in dPulv.

With this approach, we compared the amplitude of responses to stimuli delivered to the two eyes.

Before monocular deprivation, no systematic differences in eye dominance were expected; therefore, we used BOLD responses to stimuli in the two eyes for estimating the internal consistency of our results. We found that responses to the two eyes were correlated across participants in all thalamic regions (Pearson's correlation coefficients were vPulv: $r(18) = 0.58$, $p = 0.011$; LGN: $r(18) = 0.66$, $p = $

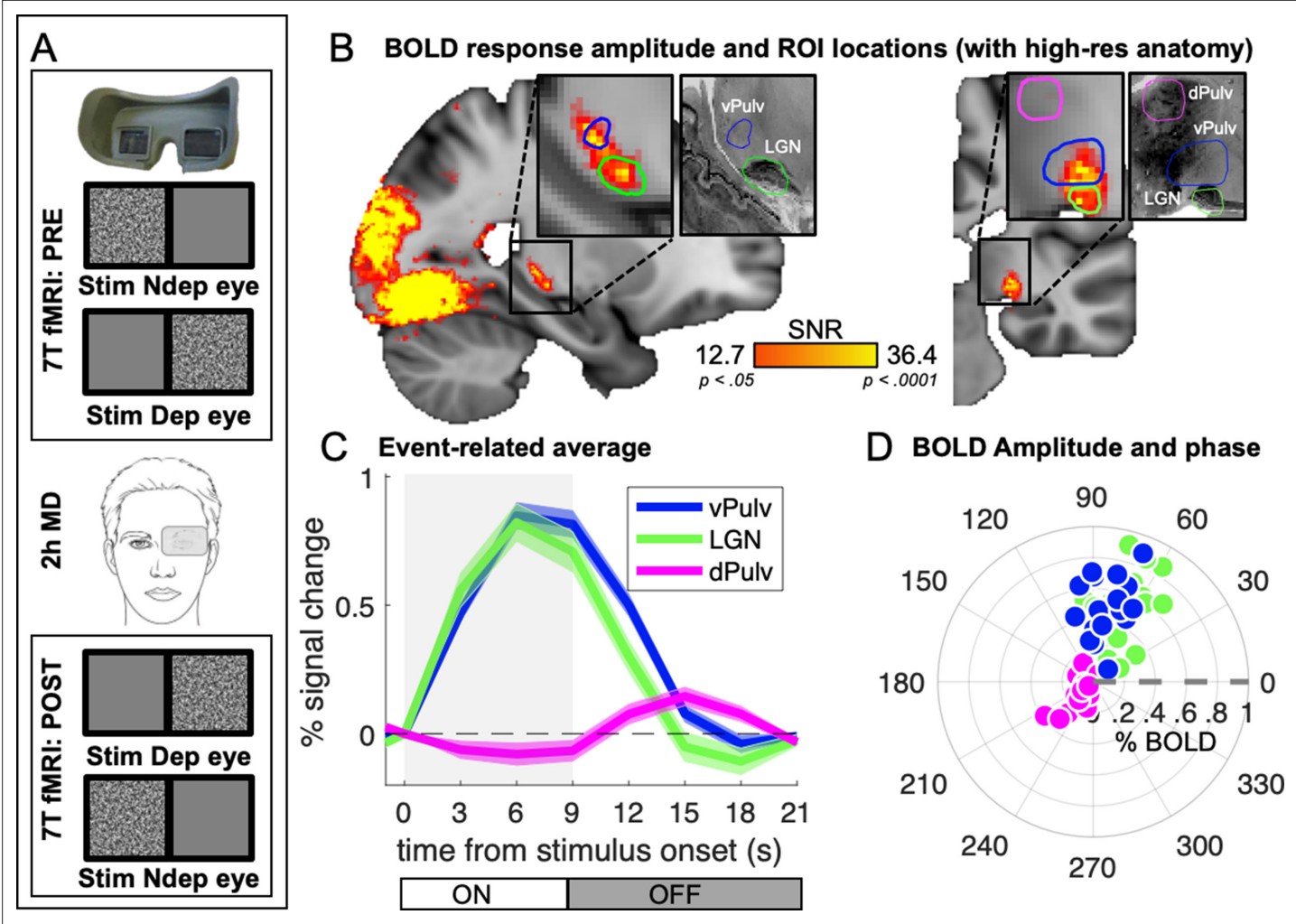

**Figure 1.** Average 7T BOLD responses in the thalamus. (**A**) Experimental design. Responses to monocular presentations of dynamic bandpass noise were recorded before and after 2 hr of monocular deprivation. Binocular rivalry was measured immediately before each scanning session and used to estimate the shift of perceptual eye dominance following deprivation. (**B**) Map of visually evoked activity, estimated by Fourier analysis of the BOLD time series, pooled across conditions and participants and mapped on the 1 mm³ MNI template. Activity in each voxel was measured as signal-to-noise ratio, defined as the amplitude at stimulus fundamental frequency divided by the mean square root of the amplitude of neighboring frequencies (as in *Biagi et al., 2015*). Maps are thresholded by the associated p values, False Discovery Rate (FDR) corrected. Colored lines outline the three independently defined subcortical regions of interest (ROIs) (*Guest et al., 2021*): ventral pulvinar (vPulv), Lateral Geniculate Nucleus (LGN), and mid-dorsal pulvinar (dPulv). For visualization purposes, these ROIs are also mapped on a publicly available high-resolution 0.4 mm³ anatomy (https://osf.io/xkqb3/; *Amunts et al., 2013*; *Xiao et al., 2019*). (**C**) Temporal dynamics of the BOLD response in the three subcortical ROIs; curves and shaded areas show means and standard errors across participants (data pooled across all sessions and averaged after subtracting the baseline BOLD signal at stimulus onset; the gray shaded area represents stimulus contrast modulation ON/OFF). (**D**) Polar plot of phase (angle) and amplitude (radius) at the stimulus fundamental frequency for each ROI. The fundamental harmonic phase of the stimulus contrast modulation corresponds to a phase of 0° (dashed gray line) and phase delays are represented as counterclockwise rotations and expressed in degrees. Each dot represents an individual participant. Averages across participants are shown in *Figure 1—figure supplement 1*.

The online version of this article includes the following figure supplement(s) for figure 1:

**Figure supplement 1.** Fourier analysis of BOLD responses.

**Figure supplement 2.** Alternative definition of thalamic regions of interest (ROIs).

0.003; dPulv: $r(18) = 0.74$, $p < 0.001$) indicating good test–retest reliability of our measurements and allowing us to examine their variations after monocular deprivation.

*Figure 2* compares response amplitudes before and after deprivation, for stimuli in the deprived and nondeprived eye. vPulv showed a significant eye by time interaction (*Figure 2A*: $F(1,17) = 14.75$, $p = 0.001$), similar as that seen in V1 (*Binda et al., 2018*). This is the hallmark of a significant short-term

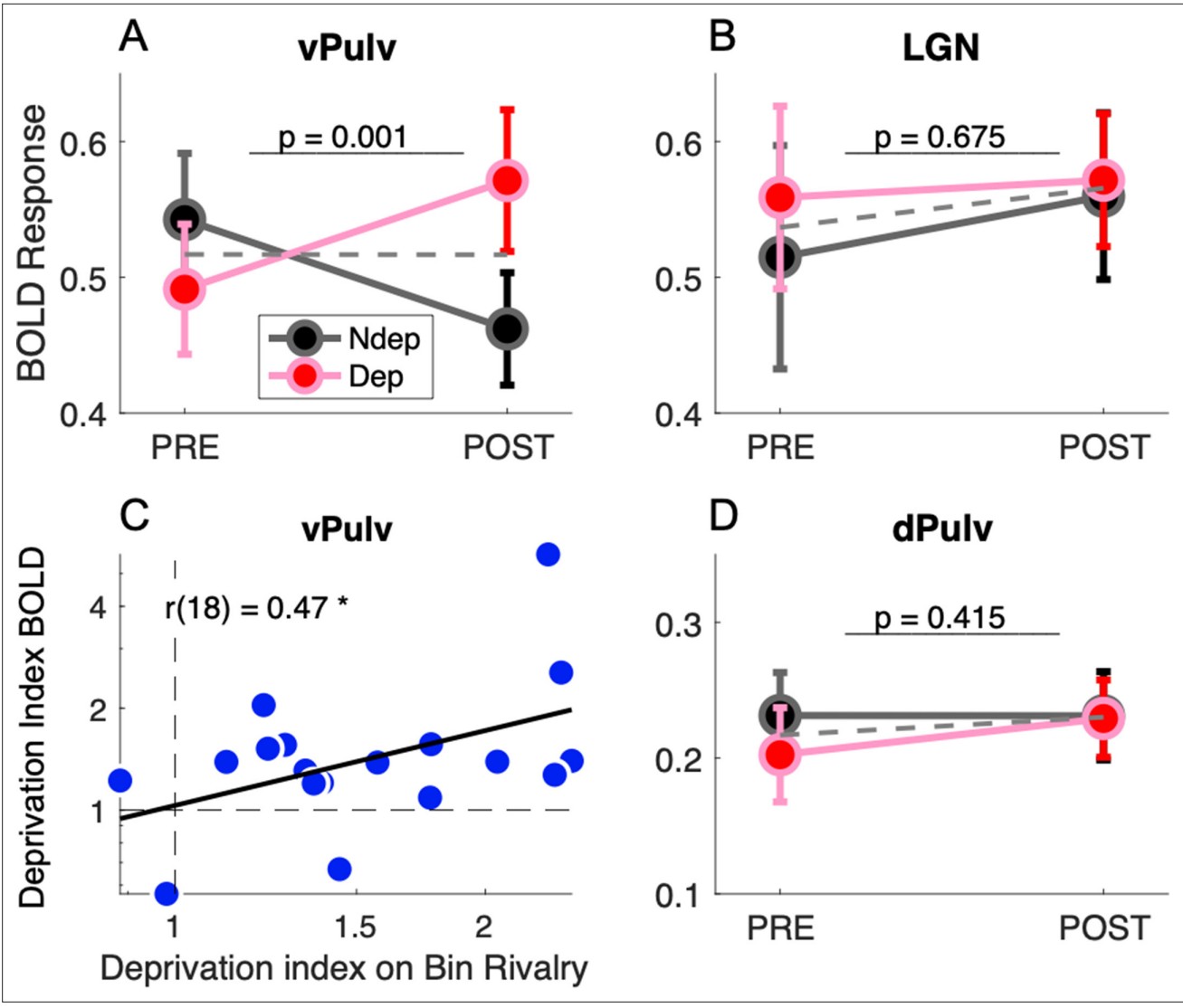

**Figure 2.** Short-term plasticity in ventral pulvinar, not in Lateral Geniculate Nucleus (LGN) or mid-dorsal pulvinar (dPulv). (**A, B, D**) Modulation of visually evoked BOLD responses with monocular deprivation, in the deprived and nondeprived eye. BOLD responses were quantified by fast Fourier analysis of the functional magnetic resonance imaging (fMRI) time series, taking the amplitude at the stimulus fundamental frequency. Symbols show means and standard error of the mean (SEM) across participants. Panels A, B, and D shows results for ventral pulvinar (vPulv), LGN, and dPulv, respectively. Note the amplified ordinate scale for dPulv data. Dashed gray lines show the average monocular responses before and after deprivation. The text inset reports the p value of the ANOVA interaction term (time by eye). (**C**) Correlation between deprivation indices computed, for each participant, for BOLD responses in vPulv and for perceptual responses during binocular rivalry (same equation as in *Binda et al., 2018*); the text inset shows Pearson's correlation coefficient, and the asterisk marks significance at p < 0.05.

The online version of this article includes the following figure supplement(s) for figure 2:

**Figure supplement 1.** Alternative region of interest (ROI) definition confirms short-term plasticity in pulvinar, not in Lateral Geniculate Nucleus (LGN).

**Figure supplement 2.** Monocular deprivation does not affect either parvocellular or magnocellular Lateral Geniculate Nucleus (LGN) divisions.

plasticity effect. In contrast, neither LGN nor dPulv showed any significant effect (*Figure 2B*, LGN, $F_{(1,17)} = 0.18$, p = 0.675; *Figure 2C*, dPulv, $F_{(1,17)} = 0.70$, p = 0.415).

We also examined response phase estimates, which did not vary across eyes or times, for any of the regions (eye by time interaction in vPulv: $F_{(1,17)} = 0.07$, p = 0.801; LGN: $F_{(1,17)} = 1.08$, p = 0.313; dPulv: $F_{(1,17)} = 0.19$, p = 0.669).

These results indicate that monocular deprivation selectively affected response amplitudes in the vPulv, but it did not reliably affect the dorsal part of the pulvinar or the LGN. The three-way interaction of factors eye, time and ROI was significant ($F_{(2,17)} = 3.45$, p = 0.039) and so was the post hoc

comparison of vPulv and LGN (p = 0.035, after Tukey–Kramer correction), implying that these thalamic regions were systematically different in their response to monocular deprivation and suggesting that the lack of significant modulations in LGN is not merely explained by lack of statistical power in this ROI.

As previously seen in V1 (**Binda et al., 2018**), we found that the interindividual variability of the plasticity effect size in vPulv was physiologically meaningful, as it correlated with the size of the behavioral effect (**Figure 2C**; $r(18) = 0.47$, p = 0.048); on the contrary, no significant correlation was found for the effect in LGN ($r(18) = −0.12$, p = 0.645, not shown) or dPulv ($r(18) = −0.04$, p = 0.867, not shown).

Together, these results suggest that the plasticity effect in the visual thalamus was selective for the vPulv region, where it correlated with the perceptual outcome of monocular deprivation.

We performed several control analyses to support these conclusions (since dPulv responded poorly to the visual stimulation, we excluded this ROI from further investigation).

First, we confirmed all our results using two alternative analyses approaches of the fMRI time series: GLM and event-related averaging. Both these methods require assumptions on the temporal dynamics of the BOLD response. GLM relies on choosing an appropriate hemodynamic response function (HRF). Using the canonical HRF previously applied to BOLD data from subcortical regions (**Koizumi et al., 2019**; **McFadyen et al., 2019**), we confirmed a reliable time by eye interaction in vPulv ($F(1,17) = 11.07$, p = 0.004), not in LGN ($F(1,17) = 0.05$, p = 0.822), the two being significantly different as testified by the significant three-way time by eye by ROI interaction ($F(1,17) = 6.89$, p = 0.018); we also confirmed that the vPulv effect correlated with the behavioral deprivation index ($r(18) = 0.51$, p = 0.031). Results were again similar when we quantified BOLD response amplitude from the event-related average curve, which we averaged in the interval between 3 and 12 s (essentially: integrating the response in **Figure 1C** over the 3–12 s interval and dividing by the duration of this interval). Again we found a reliable time by eye interaction in vPulv ($F(1,17) = 16.07$, p = 0.001), not in LGN ($F(1,17) = 0.00$, p = 0.964), with a significant three-way interaction ($F(1,17) = 4.52$, p = 0.048) and a significant correlation between the vPulv effect and the behavioral deprivation index ($r(18) = 0.49$, p = 0.039).

Second, we checked that our results were not dependent upon the specific definitions of LGN and vPulv regions that we elected to use. To this end, we redefined ROIs based on two different anatomical templates, intersected with functional activations from a separate experiment. We defined an alternative pulvinar ROI based on Najdenovska et al.'s atlas (**Najdenovska et al., 2018**), which was obtained from diffusion-weighted imaging. This label does not separate visual and nonvisual subregions of the pulvinar; we used data from an independent experiment involving a subset of our participants ($N = 9$) to isolate the visually responsive subregion. Selecting the 200 most active 1 mm$^3$ voxels from Najdenovska et al.'s pulvinar, we identified a ventral cluster that largely overlapped the vPulv region used for our main analyses (**Guest et al., 2021**; **Arcaro et al., 2015**; **Guest et al., 2021**) further validating it (**Figure 1—figure supplement 2**).

Using this alternative ROI definition, we still found a significant time by eye interaction ($F(1,17) = 11.66$, p = 0.003, **Figure 2—figure supplement 1**, panel A), confirming the reliable monocular deprivation effect in the ventral (or visual) Pulvinar.

We followed a similar strategy to obtain an alternative definition of LGN. We located it based on the histological FSL atlas (**Bürgel et al., 2006**; **Bürgel et al., 1999**) and then again analyzed the 200 most active 1 mm$^3$ voxels (**Figure 1—figure supplement 1**), thereby equating ROI size between LGN and vPulv. With this alternative definition of LGN, we still found no significant time × eye interaction in LGN ($F(1,17) = 0.00$, p = 0.979, **Figure 2—figure supplement 1**, panel B).

To further understand the lack of deprivation effects in LGN, we separately analyzed the parvo- and magnocellular divisions based on a third labeling system. Previous evidence suggests that this separation is possible with high-resolution fMRI (**Denison et al., 2014**; **Müller-Axt et al., 2021**; **Qian et al., 2020**; **Zhang et al., 2015**). We applied the separation drawn from **Müller-Axt et al., 2021**, which provided yet another definition of the LGN region (as a whole, largely overlapping the other definitions used above as shown in **Figure 2—figure supplement 2A**). **Figure 2—figure supplement 2B, C** show that there was no reliable effect of deprivation in either the parvocellular division ($F(1,17) = 1.02$, p = 0.327), nor in the smaller magnocellular division ($F(1,17) = 2.97$, p = 0.103), suggesting that the two subdivisions behaved similarly and both failed to show a systematic effect of monocular deprivation.

## Discussion

Our study is the first to show evidence for short-term plasticity in the adult human thalamus. We found that 2 hr of monocular deprivation, besides shifting ocular dominance assessed with binocular rivalry and V1 monocular BOLD responses (*Binda et al., 2018*), also affects ocular drive in a specific subregion of the visual thalamus: the vPulv.

With a series of controls, we obtained strong evidence against the possibility that this is an artifact of BOLD analyses or region labeling. We confirmed the results with three different approaches and we cross-checked them with two independent atlases, always concluding that the plasticity effect is clearly observable in vPulv.

In contrast, the adjacent dPulv and LGN regions were reliably unaffected by monocular deprivation. Considering that LGN is a complex structure with diverse morphofunctional subregions (*Ichida et al., 2014*; *Nassi and Callaway, 2009*; *Shapley, 1990*), and given prior indications that the parvocellular pathway may be preferentially subject to short-term plasticity (*Begum and Tso, 2016*; *Lunghi et al., 2013*), we also separately analyzed two LGN subdivisions corresponding to the magno- and parvocellular subregions (*Denison et al., 2014*). Both subregions lacked any detectable effect of monocular deprivation. Of course, this does not imply that LGN lacks short-term plasticity potential in humans, which may well emerge in other contexts (*Castaldi et al., 2016*; *Levine et al., 2020*), for example in pathology (*Bhat et al., 2022*; *Mikellidou et al., 2019*; *Muckli et al., 2009*) or during a stabilization of the short-term plasticity effect, as observed for repeated monocular deprivations in amblyopia (*Lunghi et al., 2019*). LGN and/or dPulv plasticity could have emerged in the context of a different task, requiring the allocation of attention to the stimuli. In the passive-viewing conditions used here, plasticity effects were strong and reliable in vPulv but undetectable in dPulv and LGN; this indicates that, if plasticity effects were present in dPulv and LGN, they were small and/or inconsistent.

Our results are coherent with neurophysiological measurements in the same three regions in behaving nonhuman primates, made in the context of a binocular rivalry task (*Wilke et al., 2009*). By measuring single cell activity, *Wilke et al., 2009* found that vPulv tracked oscillations even when stimulation was passively delivered, while dPulv tracked perception only when the animal engaged in the perceptual task; in contrast, LGN activity never followed perceptual oscillations – in contrast with evidence from human fMRI studies, where LGN responses did follow perceptual oscillations (*Haynes et al., 2005*; *Wunderlich et al., 2005*). In line with the observations by *Wilke et al., 2009*, our results confirm the close relationship between changes in visual perception and changes in vPulv activity: the short-term plasticity effect showed interindividual variability that correlated with the behavioral boost of the deprived eye measured by binocular rivalry. However, our findings extend this association beyond the context of bistable perception: we observed modulations of vPulv responses during passive monocular stimulation, not during binocular rivalry perception. We only used binocular rivalry to index shifts in ocular dominance following deprivation, to be correlated with BOLD modulations; alternative psychophysical indices (e.g., binocular fusion; *Sheynin et al., 2019*) should in principle highlight similar correlations with the changes of ocular drive in vPulv.

The lack of a reliable monocular deprivation effect in LGN suggests that adaptation at the level of monocular cells (in LGN or in the retina) is not likely to mediate the boost of the deprived eye. This negative finding suggests that the short-term plasticity effects seen in visual cortex (in the same dataset, *Binda et al., 2018*) does not result from modulations in the feedforward input to V1.

We cannot exclude that the monocular deprivation effect in vPulv originates within the small inferior subregion of the pulvinar receiving direct input from the retina and the colliculus (*Kwan et al., 2019*). However, given the small size of this subregion compared to the extent of vPulv ROI and considering that vPulv is mainly driven by binocular cortical input (*Bender, 1982*), we suggest that monocular deprivation effects are more likely inherited from the visual cortex input, although intranuclear connectivity within vPulv and cortico-pulvinar-cortex loops (*Sherman and Guillery, 2002*) may contribute to sustaining and amplifying these effects.

This hypothesis opens the question why we selectively observed the monocular deprivation effect in vPulv and not in LGN, given that both regions receive massive corticofugal input. We see two possibilities. First, it is known that corticofugal connections to LGN and vPulv generally have independent origin (*Sherman and Guillery, 2002*) that might be differently affected by the monocular deprivation. Second, in LGN, direct retinal input is expected to account for a large portion of the BOLD responses, and this could mask the modulation of cortical input following monocular deprivation. In contrast,

vPulv serves as a hub for converging signals from vast portions of the occipital and temporal cortex (*Arcaro et al., 2018*; *Kaas and Lyon, 2007*), where short-term plasticity is the strongest (*Binda et al., 2018*). If short-term plasticity effects depend on corticothalamic signals, it is reasonable to assume that these will be stronger, more stable, and ultimately easier to detect in vPulv than in LGN.

Our observation of short-term plasticity in vPulv is in line with growing evidence on the importance of the thalamus in active vision (*Saalmann and Kastner, 2011*). The traditional view of the thalamus as a passive relay of peripheral information has been overruled by evidence that thalamic neurons actively regulate information transmission to the cortex and between cortical areas (*Saalmann and Kastner, 2011*). This is particularly true for the pulvinar, which has been involved in a variety of mechanisms, including the modulation of response magnitude through gain control (*Fiebelkorn and Kastner, 2019*; *Purushothaman et al., 2012*) and synchrony of neurons (*Saalmann et al., 2012*) according to behavioral demands. These may be implemented through loops of cortico-pulvinar-cortical pathways (*Jaramillo et al., 2019*), which allow for filtering or gating incoming information. These functions have been often studied in the context of attention (*Zhou et al., 2016*) and primarily associated with dorsal subregions of the pulvinar. However, gain control of cortical responses is likely to participate in setting ocular dominance and regulating its short-term changes (*Lunghi et al., 2011*; *Spiegel et al., 2017*) and vPulv could be involved in these regulations, which could explain the correlation between BOLD modulations in this subcortical area and perceptual modulations. Interestingly, the concept that the visual pulvinar plays a fundamental role in short-term plasticity is also supported by a recent human neuroimaging study, where pulvinar was suggested to gate GABAergic inhibition in the cortex and the associated short-term learning effect (*Ziminski et al., 2021*).

In conclusion, the present study showed that short-term monocular deprivation effects, which are widespread in cortical visual areas, also extend to subcortical regions. Within the thalamus, plasticity mainly affects the ventral portion of the pulvinar – the portion of this nucleus that shows the most obvious visual functions and the strongest recurrent connections with visual cortex. Although our fMRI data do not allow to ascertain the origin of the plasticity effect, our observations open the possibility that vPulv plays a role in setting ocular dominance and maintaining its plasticity in adulthood.

## Materials and methods
### Participants and monocular deprivation procedure

Experimental procedures are in line with the declaration of Helsinki and were approved by the regional ethics committee [Comitato Etico Pediatrico Regionale—Azienda Ospedaliero-Universitaria Meyer—Firenze (FI)] and by the Italian Ministry of Health, under the protocol 'Plasticità e multimodalità delle prime aree visive: studio in risonanza magnetica a campo ultra alto (7T)' version #1 dated 11/11/2015. Written informed consent was obtained from each participant, which included consent to process and preserve the data and publish them in anonymous form. Twenty healthy volunteers with normal or corrected-to-normal visual acuity were examined (8 females and 12 males, mean age = 27 years). Sample size was set based on the minimum number of participants ($N$ = 17) required to reliably detect a medium sized correlation between MRI and psychophysical data: $r$ = 0.62 or higher, as reported in previous MR work on short-term plasticity (*Lunghi et al., 2015b*). Two (male) participants were excluded. One because of strong eye dominance (already excluded for the analyses in *Binda et al., 2018*) and the second due to a large vein passing near LGN that could bias the BOLD response. This left 18 participants (8 females and 10 males). We analyzed data from two fMRI sessions, before and after 2 hr of monocular deprivation, performed by patching the dominant eye with a translucent patch.

Binocular rivalry was measured in two short sessions (each comprising two runs of 3 min each), immediately before each fMRI session to estimate the transient ocular dominance shift (pre- vs. post-deprivation). Stimuli were presented on a 15-inch LCD monitor, placed at 57 cm distance and were viewed through anaglyph red-blue goggles (right lens blue, left lens red). Stimuli were composed of two oblique orthogonal red and blue gratings (orientation: ±45°, size: 3°, spatial frequency: 2 cpd, contrast 50%), surrounded by a white smoothed circle, presented on a black uniform background in central vision. Peak luminance of the red grating was reduced to match the peak luminance of the blue one using photometric measures. During stimulus presentation, participants were asked to respond with the computer keyboard and report which grating (red or blue or a mixture of the two) they perceived as dominant by continuous keypresses.

The effect of monocular deprivation on perception and brain activations was estimated by computing a deprivation index (*Binda et al., 2018*). This is the post- to predeprivation ratio of values '*y*' for the deprived eye, divided by the same value for the nondeprived eye:

$$DI = \left( \frac{y_{DepPOST}}{y_{DepPRE}} \right) / \left( \frac{y_{NdepPOST}}{y_{NdepPRE}} \right)$$

with '*y*' defined as binocular rivalry phase durations or BOLD responses. Using the same equation to compute the deprivation effects on psychophysical and BOLD data allowed for correlating them across participants (*Figure 2C*).

## fMRI acquisition protocols and analyses

Detailed information on the protocol and data preprocessing may be found in *Binda et al., 2018*. In our previous publication, we limited our analyses to the cortical projections of fMRI time series and focused on BOLD responses in the visual cortex. Here, we analyzed fMRI time series in the volume and focused on subcortical visual structures. Individual participants' data were aligned to a standard anatomical template, the MNI atlas, using ANTs (*Avants et al., 2008*; *Avants et al., 2011*). ANTs aligned T1 anatomical images (acquired with 1 mm isotropic resolution) to the MNI template available in FSL (*Collins et al., 1995*; *Mazziotta et al., 2001*), by means of an affine registration matrix and a warp field. These were used to transform individual participants' preprocessed BOLD data (EPI-GRE with 1.5-mm isotropic resolution and TR = 3 s, which had been slice-time, motion, and distortion corrected) to the MNI space through the antsRegistrationSyN.sh routine (*Tustison and Avants, 2013*). Purely for visualization purposes, we also mapped our ROIs to a high-resolution anatomy, downloaded from a public source (https://osf.io/xkqb3/; *Amunts et al., 2013*; *Xiao et al., 2019*), which is shown in *Figures 1 and 2*, *Figure 1—figure supplement 2* to facilitate visual appreciation of the consistency and placement of our ROI.

BOLD time series were averaged across voxels within each ROI (see below), resulting in one time series per each of the 18 participants, two ROIs and four conditions (stimulating the deprived and non-deprived eye, before and after monocular deprivation). Individual BOLD time series were transformed into percent signal change units (by subtracting and dividing by the mean signal) and detrended.

We acquired four BOLD time series per participant, two before and two after monocular deprivation. In each series, only one eye was stimulated, and the other viewed a midlevel gray image. Stimuli consisted of bandpass filtered, dynamic noise images presented in a block design, with 9-s long periods of stimulation (during which the noise stimulus was refreshed 8 times per second) separated by 12 s of interstimulus intervals (midlevel gray screen), repeated 10 times. Across blocks, the spatial frequency cutoff of the bandpass filter was varied. Unlike in *Binda et al., 2018*, here we pooled across spatial frequencies, for both theorical (spatial frequency tuning in the thalamus is not expected to be as sharp as in the cortex) and practical reasons (pooling across repetitions compensates for the lower SNR of the subcortical regions). This turned our stimulus into a periodic alternation of ON (9 s) and OFF periods (12 s), expected to generate periodic visually evoked responses, the amplitude and phase of which can be efficiently estimated with Fourier analysis (stimulus cycle was equal to 21 s or 0.047 Hz). This method is summarized in *Figure 1—figure supplement 1*. The advantage of this method is that it does not make assumptions on the latency of the response, which is captured by the phase parameter, and it is free to vary across regions.

To map visual activity across brain volumes, Fourier analyses were performed on timecourses averaged across conditions and participants. For each voxel, we calculated an SNR value by taking the Fourier amplitude at the stimulus fundamental frequency (10 cycles per scan), divided by mean square root of the amplitude at the neighboring frequencies (*Biagi et al., 2015*). SNR values were associated with p values, computed as the inverse of the associated *F*-distribution, which were corrected to threshold the maps at p < 0.05 FDR. The results are shown in *Figure 1*, *Figure 1—figure supplement 2*.

We complemented the Fourier analysis approach with two other methods.

First, we used an event-related averaging approach to estimate the profiles of fMRI responses. We selected 21 s long (7TRs) BOLD epochs following each stimulus onset and averaged across epochs (of which we had 10 per acquisition). We assumed that the response occurs between 3 and 12 s from stimulus onset, and we used the average over this interval to estimate its amplitude.

Second, we used a GLM, and we assumed a canonical (two-gamma) HRF as previously used to model subcortical responses (*Koizumi et al., 2019*; *McFadyen et al., 2019*)

$$h\left(t\right) = \frac{t^{\alpha_1 - 1}\beta_1^{\alpha_1}e^{-\beta_1 t}}{\Gamma(\alpha_1)} - c\frac{t^{\alpha_2 - 1}\beta_2^{\alpha_2}e^{-\beta_2 t}}{\Gamma(\alpha_2)}$$

where $t$ is time, $\alpha_1 = 6$, $\alpha_2 = 16$, $\beta_1 = \beta_2 = 1$, $c = 1/6$ and $\Gamma$ represents the gamma function.

We generated a stimulus predictor (boxcar function representing the stimulus ON/OFF periods, convolved by the HRF) and two nuisance predictors (a linear trend and a constant) and we extracted the corresponding beta-weights by linear regression.

## Localizer runs

Our alternative ROI labeling approach required selecting visually responsive voxels within broadly and anatomically defined thalamic regions. We performed this selection based on an independent dataset collected with a secondary experiment in a subset of our participants ($N = 9$). Using the same setup and fMRI parameters as in the main experiment, we acquired BOLD responses while a full-screen full-contrast binary-noise image (check size of 0.15°, 0.37°, 0.77°, 1.43°, and 3.3°) was refreshed 8 times per second and presented in blocks of 9 s, followed by 12-s interstimulus intervals. The sequence was repeated 10 times within a run, and in four runs per participant. As for the main experiment, our analyses ignored this stimulus dimension and simply treated timecourses as a periodic oscillation of stimulus ON and OFF periods. Responses were estimated after averaging timecourses across subjects and runs, creating one map of visually responding voxels. Visual activations were defined as SNR values computed with the same Fourier analysis approach used for the main experiment.

## ROI definition

Thalamic ROIs were defined in the MNI space based on publicly available atlases; fMRI timecourses were averaged across voxels (pooled across hemispheres) in each ROI.

ROIs for the main analysis were taken from the recently published NSD dataset, for which they were defined based on a combination of functional data (retinotopic mapping experiments) constrained with anatomical features (*Guest et al., 2021*; *Arcaro et al., 2015*). The mid-dPulv ROI was created after combining the dorsal and medial components of the pulvinar from the NSD dataset, obtaining a region of about the same size as the vPulv ROI (pilot analyses confirmed that responses in the two separate regions, medial pulvinar and dPulv, did not systematically differ).

ROIs for the confirmatory analyses were based on three additional MNI atlases. Pulvinar was labeled according to an atlas developed from diffusion tensor imaging data (*Najdenovska et al., 2018*); LGN was labelled according to an histological atlas available in FSL (*Bürgel et al., 2006*; *Bürgel et al., 1999*), setting coverage threshold to 50%. These anatomical labels were intersected with a map of visual responses from our Localizer runs (available for a subset of our participants, see above). In each hemisphere and ROI, we selected the 200 voxels with highest SNR, and we used these to define two visually responsive ROIs of equal size within the anatomically defined LGN and pulvinar regions.

Another confirmatory analysis focused on LGN and its parvo- and magnocellular subdivision. These were defined based on the high-resolution quantitative atlas of the LGN (*Müller-Axt et al., 2021*). Each subdivision was transformed from a probabilistic atlas to a binary ROI by setting coverage threshold to 60% (chosen to avoid overlap between parvo- and magnocellular labels).

## Acknowledgements

This work was supported by the European Research Council (ERC) under the European Union's Horizon 2020 research and innovation program, grant no. 801715 (PUPILTRAITS) and 832813 (GenPercept), by the Italian Ministry of University and Research under the PRIN2017 programme (grant MISMATCH) and FARE-2 (grant SMILY) and by the Italian Ministry of Health under the RC grant and the 5x1000 voluntary contributions to IRCCS Fondazione Stella Maris.

## Additional information

### Funding

| Funder | Grant reference number | Author |
|---|---|---|
| H2020 European Research Council | PUPILTRAITS | Paola Binda |
| H2020 European Research Council | GENPERCEPT | Maria Concetta Morrone |
| Ministero dell'Istruzione, dell'Università e della Ricerca | PRIN2017-MISMATCH | Paola Binda |
| Ministero dell'Istruzione, dell'Università e della Ricerca | FARE2-SMILY | Paola Binda |
| Ministero della Salute | RC and 5x1000 | Michela Tosetti Laura Biagi Jan W Kurzawski |

The funders had no role in study design, data collection, and interpretation, or the decision to submit the work for publication.

### Author contributions

Jan W Kurzawski, Data curation, Formal analysis, Investigation, Methodology, Writing - original draft, Writing - review and editing; Claudia Lunghi, Conceptualization, Investigation, Methodology, Writing - review and editing; Laura Biagi, Conceptualization, Data curation, Methodology, Writing - review and editing; Michela Tosetti, Resources, Supervision; Maria Concetta Morrone, Conceptualization, Formal analysis, Funding acquisition, Investigation, Methodology, Resources, Supervision, Validation, Writing - original draft, Writing - review and editing; Paola Binda, Conceptualization, Data curation, Formal analysis, Funding acquisition, Investigation, Methodology, Software, Supervision, Validation, Visualization, Writing - original draft, Writing - review and editing

### Author ORCIDs

Jan W Kurzawski ⬦ http://orcid.org/0000-0003-2781-1236
Claudia Lunghi ⬦ http://orcid.org/0000-0003-3811-5404
Laura Biagi ⬦ http://orcid.org/0000-0003-2159-439X
Michela Tosetti ⬦ http://orcid.org/0000-0002-2515-7560
Maria Concetta Morrone ⬦ http://orcid.org/0000-0002-1025-0316
Paola Binda ⬦ http://orcid.org/0000-0002-7200-353X

### Ethics

Experimental procedures are in line with the declaration of Helsinki and were approved by the regional ethics committee [Comitato Etico Pediatrico Regionale-Azienda Ospedaliero-Universitaria Meyer-Firenze (FI)] and by the Italian Ministry of Health, under the protocol 'Plasticità e multimodalità delle prime aree visive: studio in risonanza magnetica a campo ultra alto (7T)' version #1 dated 11/11/2015. Written informed consent was obtained from each participant, which included consent to process and preserve the data and publish them in anonymous form.

### Decision letter and Author response

Decision letter https://doi.org/10.7554/eLife.74565.sa1
Author response https://doi.org/10.7554/eLife.74565.sa2

## Additional files

### Supplementary files

• Transparent reporting form

## Data availability

The data analysed for this study are available online at the following https://doi.org/10.5281/zenodo.6457759.

The following dataset was generated:

| Author(s) | Year | Dataset title | Dataset URL | Database and Identifier |
|-----------|------|---------------|-------------|-------------------------|
| Binda P | 2021 | Short-term plasticity in the visual thalamus | https://doi.org/10.5281/zenodo.6457759 | Zenodo, 10.5281/zenodo.6457759 |

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
