## [Editor Report]

This paper documents short-term plasticity in a subcortical region, the ventral division of the pulvinar, following monocular deprivation in adult humans. This finding advances the understanding the mechanisms of short-term visual plasticity which until now has been thought to be confined to visual cortex. The results will be of interest to neuroscientists interested in brain plasticity and has potentially broad implications for the understanding of visual processing and visual disorders.

---

## [Decision Letter]

**Decision letter after peer review:**

Thank you for submitting your article "Short-term plasticity in the visual thalamus" for consideration by *eLife*. Your article has been reviewed by 2 peer reviewers, and the evaluation has been overseen by a Reviewing Editor and Joshua Gold as the Senior Editor. The reviewers have opted to remain anonymous.

Essential revisions:

The reviewers were supportive of the work, recognizing that subcortical regions may play a role in ocular dominance plasticity and that the study may contribute to the understanding the mechanisms of short-term visual plasticity. They identified a number of strengths, including clever behavioral task, sound approach to the data analysis, compelling results and clarity of writing. However, they also identified several problems that must be addressed before the manuscript can be considered for publication in *eLife*. Both reviewers felt that additional analysis is necessary to support the main conclusions of the paper and pointed out that key studies highly relevant to the work should be referenced and discussed. In the revised manuscript, address the following points:

1. Please add the analysis of the dorsal subdivision of the Pulvinar to parallel the analysis of the ventral subdivision included in the manuscript.

2. Please discuss the implications of anatomy and physiology of the LGN and vLPul in predicting and explaining the observed outcomes.

3. The question was raised whether in addition to the amplitude, phase analysis would beneficial. The need for additional details describing the analysis and explaining quantification of the effects were also noted.

The critiques below provide detailed information listing the missing references and a recommendation concerning a more effective conceptual framing of the work. Please use these comments to revise the manuscript.

*Reviewer #1 (Recommendations for the authors):*

Figure 1. Since the FFT analysis is primarily adopted in this paper, the authors may elaborate more details on this analysis. For instance, any example figure for the FFT results. What is the root mean squared error of the sinusoidal function? What is the range of p values that the activation values between 0.8-2.8 cover? Most readers are perhaps more familiar with p values and have no idea what the activation value means.

Line 343: Figure 1A should instead be Figure 1B.

*Reviewer #2 (Recommendations for the authors):*

There are some key issues with the manuscript, which I would like to see addressed:

1. There are major differences between LGN and vPulv properties and connectivity that should be discussed right up front, and which could have predicted from the outset that significant differences would be observed in these 2 subcortical nuclei. LGN relay neurons receive their strongest driving input from a single eye and are considered monocular. While there may be cross-talk between eye-specific information at the level of the LGN (because of intrinsic circuitry, cortical input, or both), this stands in stark contrast with vPulv neurons, which are largely binocular and receive their driving input from a range of visual cortical areas. A concise review of the literature on these subjects needs to be included in this manuscript and used to better define the hypotheses, or reframe the hypotheses better, as well as modulate the interpretation of results obtained.

2. A key animal study that asked whether and how binocular rivalry correlated with changes in LGN versus vPulv firing rates (Neural activity in the visual thalamus reflects perceptual suppression. Melanie Wilke, Kai-Markus Mueller, David A. Leopold. Proceedings of the National Academy of Sciences Jun 2009, 106 (23) 9465-9470; DOI: 10.1073/pnas.0900714106) appears to be of great relevance to the work described here and should be discussed. There are several aspects of that study on which I would like to see the present authors comment:

– For instance, what do the authors make of that study's conclusion, which was that vPulv activity reflects perceptual awareness, measured through a binocular rivalry task, whereas LGN activity does not? Is this relevant to the use of a binocular rivalry task in the present study?

– Given this prior study, a more in-depth discussion of how the present study advances the field is also warranted

3. Other fMRI work in humans reporting strong BOLD signal modulation in the LGN associated with periods of perceptual dominance and suppression during binocular rivalry should also be reported and discussed (Haynes JD, Deichmann R, Rees G. 2005. Eye-specific effects of binocular rivalry in the human lateral geniculate nucleus. Nature 438(7067):496-499; Wunderlich K, Schneider KA, Kastner S. 2005. Neural correlates of binocular rivalry in the human lateral geniculate nucleus. Nat Neurosci 8(11):1595-1602).

4. Please review the manuscript to ensure that causal claims are not made. The evidence provided here correlates changes in activity in the vPulv and LGN with short-term ocular dominance plasticity. It does not show whether these centers are or are not responsible for this plasticity.

---

## [Author Response]

Essential revisions:The reviewers were supportive of the work, recognizing that subcortical regions may play a role in ocular dominance plasticity and that the study may contribute to the understanding the mechanisms of short-term visual plasticity. They identified a number of strengths, including clever behavioral task, sound approach to the data analysis, compelling results and clarity of writing. However, they also identified several problems that must be addressed before the manuscript can be considered for publication in eLife. Both reviewers felt that additional analysis is necessary to support the main conclusions of the paper and pointed out that key studies highly relevant to the work should be referenced and discussed. In the revised manuscript, address the following points:1. Please add the analysis of the dorsal subdivision of the Pulvinar to parallel the analysis of the ventral subdivision included in the manuscript.2. Please discuss the implications of anatomy and physiology of the LGN and vLPul in predicting and explaining the observed outcomes.3. The question was raised whether in addition to the amplitude, phase analysis would beneficial. The need for additional details describing the analysis and explaining quantification of the effects were also noted.

Thank you very much for the opportunity to revise our manuscript. We made substantial changes in the analyses, introduction/discussion and figures and we believe that we could address all referees’ comments (as specified in the point-by-point response below). In particular:

1. We included a dorsal pulvinar ROI in our analyses. We find no significant visual response in this ROI. This is not surprising given that dorsal Pulvinar requires attention allocation to elicit a visual response, while our stimuli were presented in passive viewing conditions.

2. We added a new section of the introduction with a better account of the anatomy and physiology of these thalamic regions. We also explicitly outline possible predicted outcomes based on binocularity and contrast representation. We revised our discussion to refine the concept of a plasticity effect “originated” within the thalamus (as a monocular adaptation effect or through intra-thalamic inter-ocular connectivity, which we consider an unlikely scenario) versus an effect inherited from the cortex through cortico-fugal connections.

3. We added a supplementary figure outlining our analysis approach; we report both amplitude and phase estimates for all regions of interest. We also state that monocular deprivation did not affect response phase, which could be expected since phase mainly reflects the hemodynamic delay.

We trust that, after these revisions, our manuscript is adequate for publication in *eLife* and we would like to thank you and the referees for the helpful comments.

The critiques below provide detailed information listing the missing references and a recommendation concerning a more effective conceptual framing of the work. Please use these comments to revise the manuscript.Reviewer #1 (Recommendations for the authors):Figure 1. Since the FFT analysis is primarily adopted in this paper, the authors may elaborate more details on this analysis. For instance, any example figure for the FFT results. What is the root mean squared error of the sinusoidal function? What is the range of p values that the activation values between 0.8-2.8 cover? Most readers are perhaps more familiar with p values and have no idea what the activation value means.

We have included a support figure (Figure 1-supplement 1) to better describe the method we used to extract amplitude and phase at the stimulus fundamental frequency. We also explain the computation of Signal to Noise Ratio (defined as the ratio between amplitude at the stimulus fundamental and square root of the mean amplitude at the neighboring frequencies, following Biagi et al. 2015); in Figure 1 and in methods, we described the procedure to evaluate the p-values used to mask the brain-maps (after FDR correction) as it is common for fMRI results. We believe this improved the readability of our figures, and we thank you for your suggestions.

Line 343: Figure 1A should instead be Figure 1B.

Amended, thank you.

Reviewer #2 (Recommendations for the authors):There are some key issues with the manuscript, which I would like to see addressed:1. There are major differences between LGN and vPulv properties and connectivity that should be discussed right up front, and which could have predicted from the outset that significant differences would be observed in these 2 subcortical nuclei. LGN relay neurons receive their strongest driving input from a single eye and are considered monocular. While there may be cross-talk between eye-specific information at the level of the LGN (because of intrinsic circuitry, cortical input, or both), this stands in stark contrast with vPulv neurons, which are largely binocular and receive their driving input from a range of visual cortical areas. A concise review of the literature on these subjects needs to be included in this manuscript and used to better define the hypotheses, or reframe the hypotheses better, as well as modulate the interpretation of results obtained.

Thank you. We agree with this reviewer, and we have added a full discussion of these points, with particular reference to binocularity and representation of gain/contrast. We radically revised our introduction based on these suggestions, and we outline possible predicted outcomes for our experiment.

2. A key animal study that asked whether and how binocular rivalry correlated with changes in LGN versus vPulv firing rates (Neural activity in the visual thalamus reflects perceptual suppression. Melanie Wilke, Kai-Markus Mueller, David A. Leopold. Proceedings of the National Academy of Sciences Jun 2009, 106 (23) 9465-9470; DOI: 10.1073/pnas.0900714106) appears to be of great relevance to the work described here and should be discussed. There are several aspects of that study on which I would like to see the present authors comment:– For instance, what do the authors make of that study's conclusion, which was that vPulv activity reflects perceptual awareness, measured through a binocular rivalry task, whereas LGN activity does not? Is this relevant to the use of a binocular rivalry task in the present study?– Given this prior study, a more in-depth discussion of how the present study advances the field is also warranted3. Other fMRI work in humans reporting strong BOLD signal modulation in the LGN associated with periods of perceptual dominance and suppression during binocular rivalry should also be reported and discussed (Haynes JD, Deichmann R, Rees G. 2005. Eye-specific effects of binocular rivalry in the human lateral geniculate nucleus. Nature 438(7067):496-499; Wunderlich K, Schneider KA, Kastner S. 2005. Neural correlates of binocular rivalry in the human lateral geniculate nucleus. Nat Neurosci 8(11):1595-1602).

The close correspondence between activity in the thalamic nuclei and perceptual oscillations during binocular rivalry is relevant, thank you for highlighting this. We also stress that the effects of monocular deprivation can also be observed with other methods and that our results generalize beyond the context of binocular rivalry. The BOLD responses we analyze were elicited by passively view noise stimuli, delivered to one eye at a time, i.e. not during rivalry. We suggest that could be one of the reasons why dPulv failed to show a reliable monocular deprivation effect (the one point where our results diverge from Wilke et al.’s).

4. Please review the manuscript to ensure that causal claims are not made. The evidence provided here correlates changes in activity in the vPulv and LGN with short-term ocular dominance plasticity. It does not show whether these centers are or are not responsible for this plasticity.

We agree and we have rephrased the relevant sentences to eliminate causal claims. Thank you.